# Petermann ice shelf may not recover after a future breakup

Henning Åkesson [1,2,3 ✉], Mathieu Morlighem [4,5], Johan Nilsson[2,6], Christian Stranne [1,2] & Martin Jakobsson [1,2]

Floating ice shelves buttress inland ice and curtail grounded-ice discharge. Climate warming causes melting and ultimately breakup of ice shelves, which could escalate ocean-bound ice discharge and thereby sea-level rise. Should ice shelves collapse, it is unclear whether they could recover, even if we meet the goals of the Paris Agreement. Here, we use a numerical ice-sheet model to determine if Petermann Ice Shelf in northwest Greenland can recover from a future breakup. Our experiments suggest that post-breakup recovery of confined ice shelves like Petermann's is unlikely, unless iceberg calving is greatly reduced. Ice discharge from Petermann Glacier also remains up to 40% higher than today, even if the ocean cools below present-day temperatures. If this behaviour is not unique for Petermann, continued near-future ocean warming may push the ice shelves protecting Earth's polar ice sheets into a new retreated high-discharge state which may be exceedingly difficult to recover from.

[1] Department of Geological Sciences, Stockholm University, Stockholm, Sweden. [2] Bolin Centre for Climate Research, Stockholm University, Stockholm, Sweden. [3] Department of Geosciences, University of Oslo, Oslo, Norway. [4] Department of Earth Sciences, Dartmouth College, Hanover, NH 03755, USA. [5] Department of Earth System Science, University of California, Irvine, CA, USA. [6] Department of Meteorological Sciences, Stockholm University, Stockholm, Sweden. ✉email: henning.akesson@geo.uio.no

Since the late 1990s, ice shelves in Antarctica and Greenland have experienced thinning[1–4] and breakup[5–7], and the marine outlet glaciers feeding them have often retreated and accelerated (e.g. refs. [8,9]). Atmospheric and ocean warming is known to have played a key role in this pattern[10–13]. Glacier retreat is projected to continue over the coming decades to centuries due to sustained climate warming (e.g. refs. [14,15]). Nonetheless, the Paris Agreement aims to reduce emissions enough to limit future warming to 2 °C higher than in Earth's pre-industrial climate. This could eventually permit a return to a colder climate, reminiscent of that of the pre-industrial period. How marine outlet glaciers and their ice shelves would respond to a cooling climate has, however, been difficult to study due to a pervasive lack of data from historical analogues of such climate transitions over the recent past. We have few modern observations of outlet-glacier growth, grounding-line advance and floating ice shelves that thicken and expand. Thus, the role this behaviour plays in a cooling climate lacks an observational underpinning and remains uncertain. Documented ice advances exist in paleo-records (e.g. refs. [16,17]), but advancing glaciers generally leave fewer traces behind than those retreating because the geomorphological imprints are overridden by subsequent ice recession. Therefore, it remains unknown whether a return to the climate that prevailed in the pre-industrial period will permit ice sheets and glaciers to recover after decades to centuries of mass loss and retreat, and to what extent global sea-level rise from ice-sheet mass loss can be curtailed or even reversed. Specifically, it is unclear if a climatic reversal in a post-Paris world allows glaciers to re-advance, ice shelves to regrow and sea-level rise to be kept moderate.

To tackle this global question, we investigate Petermann Glacier in northwest Greenland as an example (Fig. 1a), using the Ice-sheet and Sea-level System Model (ISSM;[18]). We examine the climate conditions required for Petermann to recover to its present-day (2008) state after an ice-shelf breakup and grounding-line retreat[19]. Petermann is the largest glacier by area in northern Greenland, drains about 4% of the entire Greenland Ice Sheet by area[20,21] and has one of the northern hemisphere's few remaining ice shelves. Note that Petermann's ice shelf is confined in a fjord (sometimes referred to as an ice tongue), in contrast to many topographically unconfined ice shelves in Antarctica. Over the last decade, Petermann has lost ~40% of its ice shelf[22,23] and concerns have been raised that further breakup would lead to grounded-ice speedup, retreat and accelerated mass loss[19,24–26]. Similar scenarios have been put forward for ice-shelf-glacier systems in West Antarctica in response to ongoing and future ocean warming (e.g. refs. [27,28]).

Here we show that once a confined ice shelf has been lost, the glacier feeding it may get locked into a new stable regime of sustained mass loss even if the climate cools again. The numerical experiments we present for Petermann Glacier demonstrate that a cooling well below present-day conditions, in conjunction with sea-ice growth, is required in order to escape this high-discharge state and thus avoid considerable sea-level rise.

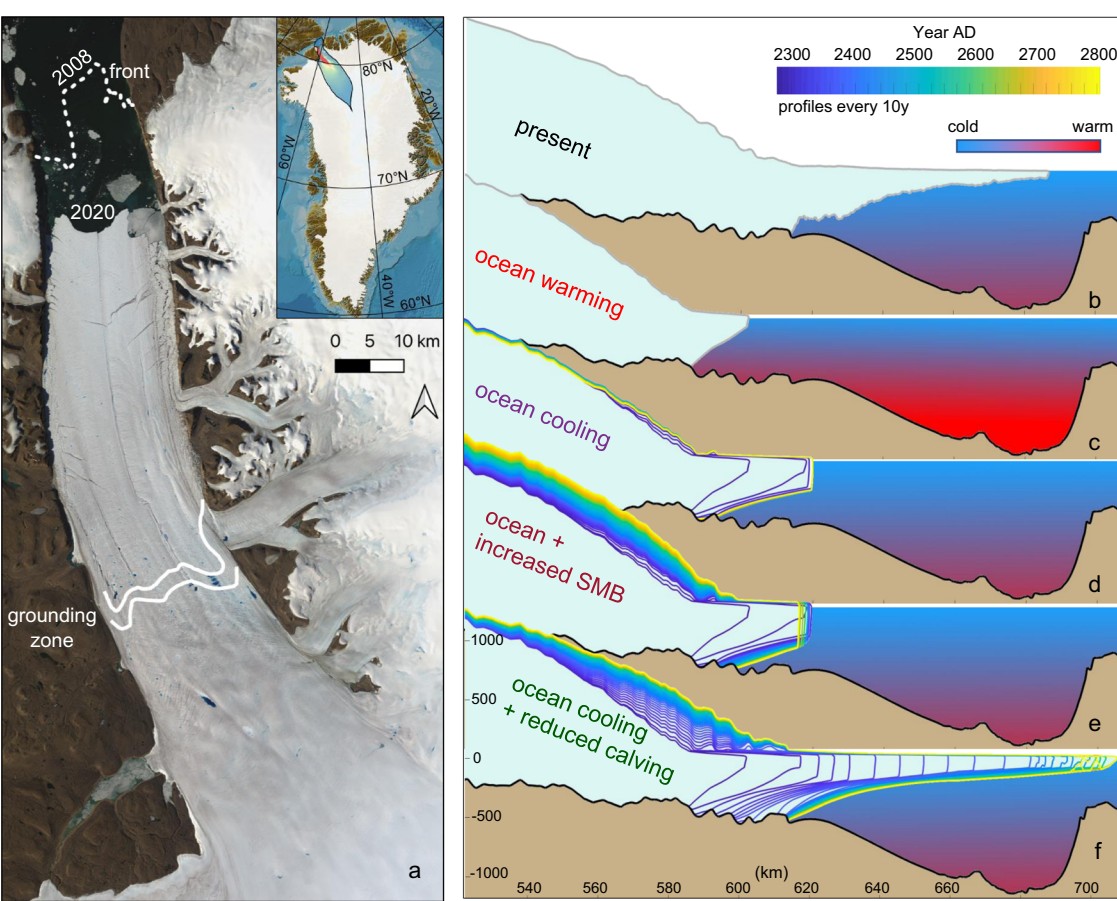

**Fig. 1 An ocean cooling and reduced calving permit Petermann's ice shelf to recover once it has disappeared. a** Present-day Petermann Glacier, **b** contemporary profile view, **c** geometry by 2300 AD after a 2 °C ocean warming. Glacier evolution in response to an attempted recovery is shown for **d** an ocean cooling, **e** an ocean cooling and an increased surface mass balance (SMB) and **f** an ocean cooling and reduced calving rates. The present-day grounding zone and calving front is shown in **a**. The satellite image is from Landsat-8, taken July 24, 2020, courtesy of the U.S. Geological Survey.

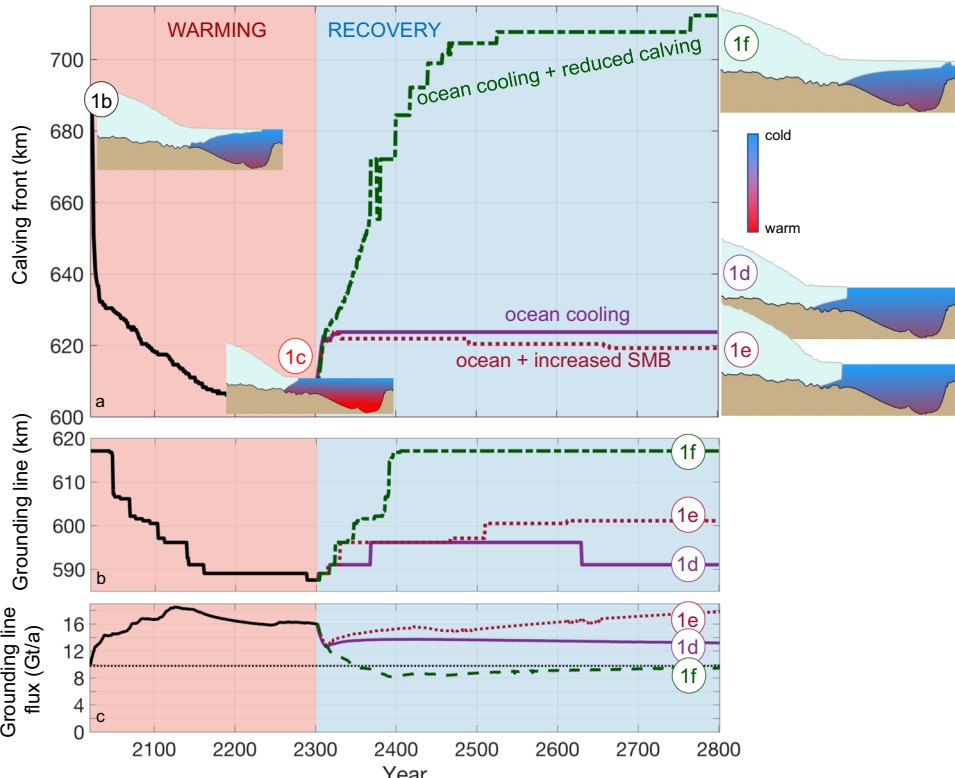

**Fig. 2 Response of Petermann Glacier Ice Shelf to future climate warming and subsequent cooling pathways.** Evolution of Petermann's **a** ice-shelf calving front, **b** grounding line and **c** grounding-line flux, for the forcings compared in Fig. 1 during the 2 °C ocean warming[19] and recovery experiments. An ocean cooling alone does not lead to recovery, neither does ocean cooling + increased surface mass balance (SMB). In contrast, an ocean cooling and reduced calving allow the ice shelf to recover. In **c**, the dashed line represents the present-day observed flux[36]. The labels (1b–1f) refer to the panels in Fig. 1. Modelled glacier geometries and schematic ocean temperatures are shown in **a** for each associated state.

## Results

**Ice-shelf breakup may cement pervasive sea-level rise.** We start all our Petermann recovery simulations from a future retreated model state (year 2300 AD; Fig 1c), which is attained as a response to a 2 °C ocean-only warming relative to present-day[19]. This corresponds to a ~3 °C ocean warming relative to pre-industrial times (before c. 1850). In a more complete future scenario, atmospheric warming would cause some additional mass loss, but this is ignored here for simplicity. Moreover, the ocean-induced melt is responsible for ~80% of Petermann's current mass loss[29]. The applied ocean forcing is based on observations and results from the ocean model MITgcm (see Section "Ocean forcing"[29,30]).

The instantaneously increased future ocean temperature ramps up the basal melt, which causes the calving front to retreat rapidly, reaching a retreated new equilibrium position after about 200 years (approximately year 2200; Fig. 2a). As the ice shelf shortens, its buttressing of grounded ice weakens due to decreased fjord-wall drag. This allows increased mass loss across the grounding line (Fig. 2c). The new future equilibrium state is thus characterised by reduced buttressing, and the ice shelf loses mass primarily through calving (Fig. S10b), rather than through basal melting as for the present-day state.

In the first suite of ice-shelf recovery experiments, we reverse the future 2°C ocean warming that caused the retreat displayed in Fig. 2a (see ref. [19] for details), thus lowering ice-shelf basal melt rates back to those associated with present-day ocean conditions (Section "Ocean forcing"; Fig. S2). If the Paris Agreement is fulfilled and the current global warming trend is reversed, such ocean cooling could occur in the decades to centuries that follow. Our experiments suggest, however, that this substantial ocean

cooling alone would not cause Petermann's ice shelf to recover (Figs. 1d and 3b). Despite the ocean temperature being returned to its present-day value, the grounding line and ice-shelf front of Petermann advance only ~3 and ~15 km, respectively (Fig. 2a, b). For context, the contemporary pre-2010 ice shelf was around 70 km long, while the modelled pre-recovery shelf is ~20 km (Fig. 1a, b). This reversal of future ocean warming will also keep ice discharge 40% higher than today and thus maintain a long-term positive contribution to sea-level rise (Fig. 2c).

These key results are insensitive to the time scales of ocean cooling; recovery fails regardless of whether the ocean cools from one year to the next, or over several centuries (Fig. S9b-d). Even completely turning off ocean-induced melt, a highly unrealistic scenario, does not allow Petermann to recover and does not prevent a lasting high sea-level contribution (Table S1). This firmly illustrates that future ice-shelf breakup and grounding-line retreat may push marine outlet glaciers into a new dynamic state with reduced ice-shelf buttressing, and higher grounding-line flux and iceberg calving rates[31]. Notably, Petermann Glacier remains in this retreated state even when the ocean temperature is returned to present-day values, which suggests that the glacier has multiple equilibrium states[32].

**Recovery in a post-Paris climate.** The dynamics of ice shelves depends on local processes, such as basal melt, but also on non-local processes such as the mass balance of the inland grounded ice[33]. We therefore examine whether reversed ocean temperature in combination with increased net surface mass balance can allow regrowth of Petermann Glacier Ice Shelf. In these experiments, we impose a positive perturbation to the model's contemporary

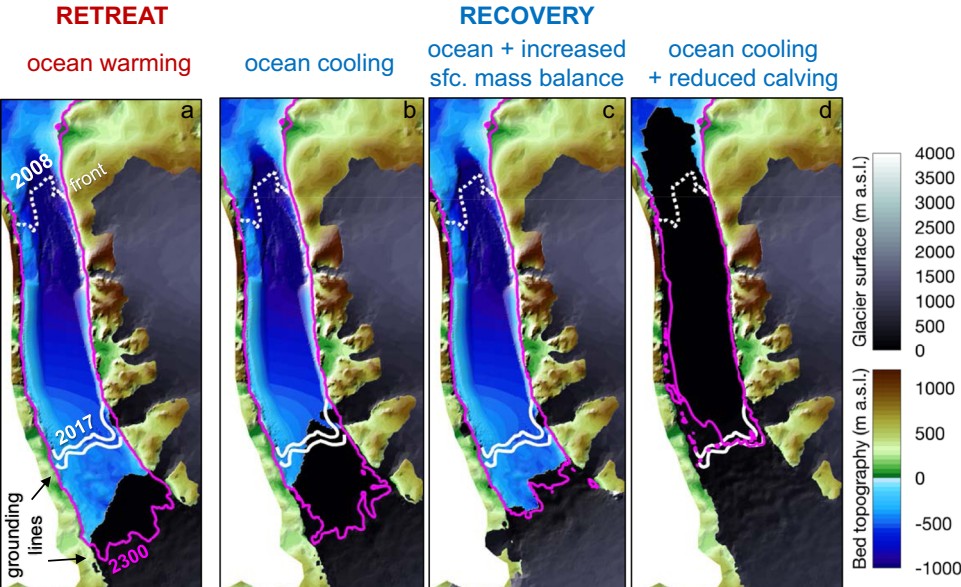

**Fig. 3 It is difficult to recover an ice shelf once it has disappeared. a** Future retreated[19] state of Petermann Glacier; panels **b**–**d** show the evolution of glacier geometry for the forcings compared in Figs. 1 and 2. Only an ocean cooling combined with reduced calving rates, as shown in **d**, allow Petermann to re-advance to its contemporary state. **b** ocean cooling and **c** ocean cooling + increased surface (sfc.) mass balance lead to continued elevated ice discharge (Fig 2c), and thus a sustained high contribution to sea-level rise even in a cool climate.

surface mass balance, where the latter is based on a regional climate model (see Section "Atmospheric forcing"[34]).

Despite these more favourable conditions applied for glacier growth, Petermann Glacier Ice Shelf does not regrow to its original extent (Figs. 1e and 3c). Whereas Petermann undergoes upstream thickening of several hundreds of meters due to the more positive surface mass balance, this thickening does not translate into grounding-line and calving-front advance (Figs. 1e and 2a, b). Instead, the increased upstream accumulation steepens the ice-stream surface and allows ice to flow twice as fast (~2000 m/a) compared to present-day. This in turn causes a steadily rising grounding-line flux and thus an accelerating contribution to sea-level rise (Fig. 2c).

**Reduced calving as an escape route from pervasive mass loss**. Thus far we have seen that a future reversal of ocean warming is not sufficient to recover Petermann's ice shelf after a future breakup (Section "Ice-shelf breakup may cement pervasive sea-level rise"), neither is a more positive surface mass balance over the glacier (Section "Recovery in a post-Paris climate"). Cooling in a post-Paris world may, however, also have other impacts on the coupled ice-sheet-ocean-climate system, for example through changes to iceberg calving (Section "Sea-ice induced advance of ice shelves"). In a suite of simulations, we consider whether reduced calving rates can aid the recovery of Petermann Glacier Ice Shelf.

In the numerical model, we impose changes to the calving regime by increasing the stress threshold in the calving law (see Section "Calving parameterisation"[35]). We hypothesise that suppressed calving is linked to sea-ice growth and increased ice mélange in Petermann Fjord. The theoretical and empirical support for this link is discussed in detail in Section "Sea-ice induced advance of ice shelves" below.

We find that reduced calving rates alone do not yield ice-shelf recovery (Figs. S10e and S11). Instead, our experiments suggest that a combined reversal of ocean warming and less vigorous calving is the only escape route from future pervasive mass loss

(Figs. 2c and S11). When the ocean cools and calving is reduced in the model, Petermann Glacier Ice Shelf can be re-established (Fig. 1f), the grounding line re-advances beyond its present-day location (Fig. 2b), and the grounding-line flux declines to its contemporary values (Fig. 2c; ~9.8 Gt/a;[36]).

Clearly ocean cooling (Section "Ice-shelf breakup may cement pervasive sea-level rise") and reduced calving are mutually dependent to trigger regrowth. Additional experiments confirm that this holds regardless of whether ocean cooling is rapid or sluggish, and whether the calving regime changes like a flip-switch or over several centuries (Fig. S9). In addition, the requirements for recovery hold both for an annual calving forcing, as presented here (Section "Calving parameterisation"), as well as when introducing calving seasonality (Section S1.1).

Note that ocean cooling and sea-ice growth do not necessarily vary in tandem in the real world. An ocean cooling in our experiments means that sub-ice shelf melt is reduced (cf. Section "Ocean forcing"). In reality, sub-shelf melt is mainly driven by available subsurface ocean heat which, in this case, is controlled by remote oceanographic conditions rather than local forcing. In contrast, local atmospheric conditions govern surface waters and sea-ice conditions. Indeed, the ocean—sea ice—atmosphere is a coupled system, but changes are likely to be asynchronous and occur on different time scales.

## Discussion

**A double-headed ice-shelf regime**. Our model study of Petermann Glacier indicates that for present-day climatic conditions, its ice shelf has two different equilibrium configurations: (1) a contemporary long ice-shelf state (Fig. 1a, b) with moderate ice discharge (Figs. 2c and S3), and (2) a short ice-shelf state, with higher grounding-line flux (Figs. 1c and 2c). Further, the numerical simulations show that both a decrease in basal melting and iceberg calving are needed to recover the ice shelf from the short to the long state (Figs. 1f, 2a and 3d). This transition has a non-linear and step-like dependence on the calving stress threshold $\sigma_{max}$ (cf. Eq. (2)): for $\sigma_{max}$ above 440 kPa (47% higher

than the present-day value of 300 kPa), the ice shelf grows and attains the long equilibrium state; for smaller values of $\sigma_{max}$, the ice shelf remains in the short state, regardless of the amount of ocean cooling. Provided sufficient ocean cooling, even stress changes of less than 1% near the critical $\sigma_{max}$ stress of around 440 kPa can trigger a feedback loop of thickening and advance, causing Petermann's ice-shelf front to advance by 70 km and its grounding line by 20 km (Fig. 2a, b and S8). In contrast, without any ocean cooling, the long- and short-shelf states are both essentially insensitive to the calving stress threshold $\sigma_{max}$. For example, once the ice shelf has reached the short regime, recovery does not occur even if $\sigma_{max}$ is increased to several times the present-day value (Calving-only experiments in Table S1).

To summarise, a transition from a short-shelf to a long-shelf regime can only occur if the shelf is preconditioned by sufficient ocean cooling. In this case, a relatively small increase in the calving stress threshold is required to trigger complete advance.

Overall, we suggest that the major control of recovery is the ice-shelf dynamics and calving regime, but with local impacts of the topography. The fjord is fairly straight so we do not expect width variations[37,38] to be influential. Upstream of the present-day grounding line there are local bedrock highs that pin the grounding line in the short-shelf regime (see Figs. 1c and S3c). The bathymetry under the present-day ice shelf is seaward-sloping yet poorly known. An outer bathymetric sill with a landward overdeepening is common seafloor morphology in fjords and not unique for Petermann. Grounding-line advance along a seaward-sloping bed requires a combination of increased upstream ice supply (dynamic and/or climatic) and ice-shelf thickening (dynamic and/or oceanic). These requirements may partly explain why advance is so difficult to attain.

The outer parts of Petermann's ice shelf[24], as well as the 79N ice shelf in northeast Greenland[39], have been interpreted as dynamically passive, providing little buttressing for upstream ice. These model studies did, however, not include transient effects and involved feedbacks, and thus excluded explicit treatment of calving or sub-shelf melt. In the more comprehensive study presented here, we show that if Petermann's ice shelf is lost beyond the passive ice limit[19], recovery is unlikely unless calving rates are reduced.

**Geological support for past Petermann regime transitions**. Our recovery experiments (Sections "Ice-shelf breakup may cement pervasive sea-level rise", "Recovery in a post-Paris climate", "Reduced calving as an escape route from pervasive mass loss") provide a mechanistic framework for the requirements to recover an ice shelf in the future. This framework may also help us to understand (re)growth of ice shelves in the past. This is relevant for the response to any climatic cooling throughout the geological history, not only in a future post-Paris climate scenario.

Geological evidence suggests that Petermann's grounding line retreated rapidly across a bathymetric sill and into Petermann Fjord during the early Holocene (c. 8700–7600 yrs BP;[40]). This was followed by ice-shelf breakup c. 6900 yrs BP, according to sediment records taken below the present-day ice shelf[16]. The ice shelf was absent for the following several thousand years and started to grow again c. 2100 yrs BP, to reach its contemporary configuration ~600 years ago (1400 AD;[16]).

Our simulations imply that a plausible explanation for the historic re-establishment of Petermann Glacier Ice Shelf was an ocean cooling combined with reduced calving. In light of empirical reconstructions[16,41] and our modelling results, the following scenario seems plausible: (1a) Before 2100 yrs BP, a relatively warm ocean and vigorous calving regime prevented ice-shelf growth in Petermann Fjord; (1b) after 2100 yrs BP, an ocean

cooling occurs, which in theory allows for an extensive ice shelf. Due to the difficulty to trigger recovery with an ocean cooling alone (Figs. 1c and 2a, b), regrowth of the ice shelf is, however, marginal; (2) at 1400 years BP a regime shift towards modern conditions occurs, with landfast sea ice in Nares Strait (see Section "Sea-ice induced advance of ice shelves"[41]); (3) a sustained weakened oceanic heat flux[42] favours sea-ice growth, dampens calving and allows for regrowth of the ice shelf to present-day conditions at c. 600 years BP (cf. Fig. 1; ref. [16]). In Section "Sea-ice induced advance of ice shelves" below we discuss the physical mechanisms involved between a cooling ocean, sea-ice expansion and reduced iceberg calving in more detail.

**Sea-ice induced advance of ice shelves**. Changes to the calving regime are clearly necessary, yet on their own not sufficient, for the ice shelf to regrow in our simulations (Fig. 1f) and thus avoid persistent mass loss (Figs. 2c and S11). We postulate that the 'ocean cooling + reduced calving' recovery (Figs. 1f, 2 and 3d) can be induced by sea-ice growth and changes to the ice mélange strength in Petermann Fjord.

*Ice mélange* is a dense mix of calved icebergs and sea ice that covers the inner part of many Greenland fjords (e.g. ref. [43], preventing the export of further calved ice out of the fjord. Mélange tends to be more prevalent and extensive in front of glaciers with vigorous calving, which usually occurs in warmer fjords. Conversely, in fjords with cooler ocean and atmospheric temperatures, calving rates are expected to be lower, but calved icebergs take longer to melt, favouring a longer-lasting mélange. In winter, landfast sea ice acts as an effective glue that strengthens the mélange[44]. A strong ice mélange has been found to suppress calving at other Greenland glaciers such as Jakobshavn Isbræ[43,45–47], by mechanically restricting iceberg calving both with and without a binding sea ice[44,48]. The critical stress threshold in our calving model ($\sigma_{max}$ in Eq. (2)) can be viewed as a proxy for mélange-strength.

When calving rates increase in a warmer climate, more icebergs are available that can be fused into an ice mélange that is strong enough to suppress calving. This has been proposed as potential dampening feedback for glacier retreat[49]. In the cooling climates considered here, such as the late-Holocene and potentially a post-Paris world, the end effect is the same, but the mechanism is different. Landfast sea-ice growth would prolong the quiet calving season in winter as well as a favour a stronger mélange. This would reduce net annual mass loss from calving and thus allow for ice-shelf advance as seen in our Petermann experiments (Figs. 1f and 2a).

For Petermann, our findings suggest that in a cooling climate, ice-mélange-induced suppression of calving can facilitate ice-shelf recovery, an effect which in our simulations is promoted by a higher calving stress threshold (Sections "Reduced calving as an escape route from pervasive mass loss" and "Calving parameterisation"). We postulate that shorter calving seasons and a stronger ice mélange, both promoted by sea-ice growth, are key physical mechanisms for ice-shelf recovery that has occurred in the past and may occur in the future. This hypothesis is supported by reconstructions of sea-ice and ocean conditions over the Holocene. Based on marine sediment cores and multiple sea-ice proxies, Detlef et al.[41] suggest that changes in sea-ice conditions in the Nares Strait offshore of Petermann Fjord coincide with the reconstructed regrowth of Petermann Glacier Ice Shelf during the late Holocene (see Section "Geological support for past Petermann regime transitions" above; ref. [16]). In particular, they find that the sea-ice regime in Petermann Fjord was discontinuous during the warm mid-Holocene, with prolonged summers with open water, consistent with a weak ice mélange and the

absence of an ice shelf. This was followed by a shift towards more extensive, near-perennial sea ice from 3,900 yrs BP onwards, and a shift from a northern to a southern ice arch in the Nares Strait. This precedes the reconstructed ice-shelf recovery at 2100 yrs BP[16]. These changes may have been accompanied by an ocean cooling similar to what we impose in our experiments. This connection between ocean and sea ice is seen also under modern conditions, as sea ice modifies the ocean circulation in Nares Strait[41,42]. Specifically, landfast sea ice weakens the inflow of warm Atlantic Water into Petermann Fjord[42], while more mobile sea ice is associated with increased warm-water inflow. Detlef et al.[41] also suggest that an earlier-than-usual breakup of landfast sea ice in spring preceded the major calving events at Petermann in 2010 and 2012.

Increased sea ice offshore of Petermann Fjord may also have prevented flushing of the ice mélange, otherwise done efficiently by prevailing katabatic winds. This sea-ice damming effect is currently visible in Sherard Osborn Fjord, located north of Petermann Fjord, where Ryder Glacier drains. The thick sea ice in Lincoln Sea offshore of Ryder is pushed towards the fjord entrance and thereby efficiently prevents iceberg export from Ryder Glacier[50].

**Study limitations and future work.** The difficulty in recovering Petermann Glacier after the breakup of its ice shelf is robust over multiple time scales and forcings, as discussed above. There are nonetheless several aspects of our study that must be considered when evaluating the results. We have parameterised some key physical processes, including iceberg calving (Section "Calving parameterisation") and basal friction (Section "Ice-flow model"; Equation (1)). The type of calving law used can impact grounding-line dynamics and ice-shelf stability (e.g. ref. [51]). Still, the calving law we employ has been compared against several others for a range of Greenland glaciers[52], and was found to give the most realistic behaviour. Similarly, the choice of friction law influences glacier flow, grounding-line behaviour and the response to external forcing[19,53–56]. Nevertheless, we find that post-breakup recovery of Petermann is similarly difficult using a Schoof friction law (Section S1.2; Figs. S5 and S6; refs. [57,58]).

Future studies could consider more elaborate parameterisations of the ocean (e.g. refs. [59,60]) and atmospheric forcing than what we use (Sections "Ocean forcing" and "atmospheric forcing"). The atmospheric forcing is relatively simple; to impose a climate cooling (Section "Recovery in a post-Paris climate"), we shifted the present-day surface mass balance field in a positive direction (Section "Atmospheric forcing"). An alternative would be to exploit surface mass balance products from model intercomparisons such as ISMIP6[14], combined with a regional surface mass balance model. Nevertheless, recovery triggered from the atmospheric side appears very difficult; even with stronger positive shifts to the surface mass balance than what is presented in Figs. 1e and 3c, recovery still fails (Table S1).

In a warmer climate, surface ablation will strengthen and runoff will thus increase. This supraglacial water can accumulate as surface lakes at an ice-shelf surface. These lakes can drain rapidly to the ice-shelf base through large-scale *hydrofracture*, which has been suggested as the trigger of the disintegration of some Antarctic ice shelves (e.g. refs. [6,61]). Supraglacial lakes cover ~2.8% of the surface of Petermann's contemporary ice shelf[62], but most of these lakes drain or are evacuated through supraglacial rivers every summer (cf. ref. [63]). In the present climate, Petermann Glacier Ice Shelf is therefore not thought to be susceptible to collapse due to large-scale hydrofracture. In theory this could change with the increasing availability of supraglacial water in a warmer climate[62]. In the case of the cooling climate

considered here, the reduced availability of surface water may reduce fracture development. This would mean a higher effective viscosity of the ice shelf and thus increased buttressing (cf. ref. [64]), in turn potentially reducing calving rates, providing another link between the atmosphere and ice-shelf advance.

We postulate that sea-ice growth and a stronger ice mélange is a viable escape route from sustained post-breakup sea-level rise. Empirical glacier reconstructions support this idea (Section "Geological support for past Petermann regime transitions"), and our calving parameterisation implicitly accounts for the effect of ice mélange through the calving stress threshold (Section "Calving parameterisation"). Still, this hypothesis is yet to be tested in a model framework with an explicit parameterisation of ice mélange[49,65].

Finally, we have illustrated that a major climatic cooling would be needed to regrow Petermann Glacier Ice Shelf after its breakup. These findings may be transferable to topographically less confined Antarctic ice shelves, with implications for the future Antarctic contribution to sea-level rise. The general notion is that the laterally extensive, unconfined ice shelves typical for present-day Antarctica provide little buttressing to upstream ice[64,66]. However, the presence of sea ice may increase the effective viscosity of an ice shelf, allowing the shelf to buttress ~10% of the extensional driving stress[64]. Similarly, reduced sea ice offshore of Antarctic ice shelves, as well as loss of protective landfast ice, has been shown to coincide with shelf disintegration[67]. Conversely, heavy pack ice (ice mélange) has been suggested to protect ice-shelf fronts from flexure by ocean swells, reducing calving rates and preventing collapse[67], and even limiting the effects of a potential marine ice-cliff instability[65]. While intriguing and physically sound, a direct causal connection between sea-ice loss and shelf breakup is yet to be established, or similarly, between the sea-ice expansion and shelf advance postulated here. Deciphering these aspects remain a priority for future work.

## Summary and outlook

We have used a numerical ice-sheet model to study the recovery of Petermann Glacier in northwest Greenland, after potential future ice-shelf breakup. Our model simulations show that post-breakup recovery of the ice shelf is difficult and requires major climatic cooling to occur. For Petermann, a mere reversal of future ocean warming back to contemporary conditions is insufficient for recovery, and dynamic ice discharge remains 40% higher than the present. Instead, we highlight sea-ice and ice mélange-induced suppression of calving, potentially accompanying colder ocean temperatures, as a viable escape route from sustained mass loss and associated sea-level rise. While the suppressive effect of sea ice on calving is supported by empirical evidence, it is not explicitly modelled and only implicitly accounted for in our calving parameterisation. Given the difficulty to regrow ice shelves once they have collapsed, the rationale to avoid ice-shelf breakup in the first place should be clearer than ever. Future research needs to pin down the exact mechanisms and thresholds of ice-shelf breakup, and to what extent ice shelves in Antarctica exhibit the same behaviour as Petermann Glacier Ice Shelf in Greenland.

## Methods

**Ice-flow model.** Ice dynamics is modelled using the two-dimensional Shelfy-Stream Approximation[68,69] on a finite-element mesh comprising 42,000 elements. The mesh resolution varies from 0.5 to 10 km based on the steepness of the bed topography. In addition, the mesh is refined to 0.5 km where observed velocities exceed 500 m/a.

The model domain (Fig. S1) includes Petermann's drainage basin as delineated based on observed ice-surface velocities[70,71]. The domain extends ~70 km offshore of the present-day calving front, to facilitate glacier advance during the recovery

experiments outlined in Sections "Ice-shelf breakup may cement pervasive sea-level rise", "Recovery in a post-Paris climate", "Reduced calving as an escape route from pervasive mass loss". Bedrock topography is taken from BedMachine v3[72] and we use ice-sheet geometry from the Greenland Ice Sheet Mapping Project[73–75].

Grounding-line migration is modelled with subelement resolution and is based on a flotation criterion[76]. The model time step is 0.05 years (18.25 days).

Basal friction is modelled using a linear viscous Budd law[77], which calculates basal drag $\tau_b$ as

$$\tau_{\mathbf{b}} = -\alpha^2 N \mathbf{u}_b, \tag{1}$$

where $\alpha$ is a friction parameter, $\mathbf{u}_b$ is basal velocity and $N$ the effective pressure. We assume a perfect hydrological connection between the subglacial drainage system and the ocean, defining $N$ as the difference between ice overburden and hydrostatic pressure: $N = \rho_i g H + \rho_w g z_b$, where $\rho_i$, $g$, $H$, $\rho_w$, and $z_b$ are ice density, gravitational acceleration, ice thickness, seawater density and bed elevation, respectively. While this parameterisation of the effective pressure is not realistic far from the coast, it allows for a smooth transition of basal stress in the grounding zone, which has been shown to significantly improve the fit with remote sensing data[56]. The basal friction parameter $\alpha$ is inverted for using an adjoint method and remotely sensed ice velocities[70,71].

Similarly, ice-shelf viscosity is estimated using an adjoint method minimising the absolute misfit between modelled and observed velocities (see ref. [19] for details). Ice viscosity for grounded ice is assumed to be uniform and corresponds to an ice temperature of −12 °C based on Greenland-wide ISMIP6 experiments[14].

**Calving parameterisation.** A von Mises law[35] is used to simulate calving-front evolution, where the calving rate $c$ depends on the tensile stress:

$$c = |\mathbf{u}| \frac{\tilde{\sigma}}{\sigma_{\max}}, \tag{2}$$

where $\tilde{\sigma}$ is the von Mises tensile stress, which depends only on the tensile strain rate (see ref. [35] for details), and $\sigma_{max}$ is a stress threshold. We use $\sigma_{max} = 1$ MPa for grounded ice, and calibrate $\sigma_{max}$ to 300 kPa for floating ice by reproducing the present-day ice-shelf margin after a 50-year transient relaxation, where the grounding line and calving front can evolve freely (see ref. [19] for details). Floating-ice thickness smaller than 100 m is not allowed, to avoid unrealistic calving model behaviour along the fjord walls.

**Climate forcing**

*Atmospheric forcing.* Present-day atmospheric forcing is imposed using a climatic surface mass balance (1979–2014) from the regional climate model MAR 3.5.2[34]. This surface mass balance is derived from monthly means for each year.

In a colder climate, decreased ablation is not expected to be compensated by an equivalent increase in accumulation[78]. This means that coastal areas would thicken more than would the interior ice sheet, which results in a flatter ice-surface profile. For example, model studies and empirical evidence suggest that a divide higher than ~3500 m a.s.l. is unlikely to have occurred during past cold periods, such as the Last Glacial Maximum and the Younger Dryas[79–81]. In comparison, the elevation of the present-day divide is only a few hundred meters lower[74]. In simulations where positive shifts of the surface mass balance are imposed (ocean cooling + increased SMB), see Section "Recovery in a post-Paris climate" and Table S1, we thus apply elevation-dependent anomalies to the climatic surface mass balance. These anomalies decrease linearly from sea level to the maximum ice-surface elevation.

The elevation-dependency factor $\Gamma$ for these surface mass balance anomalies is calculated as

$$\Gamma = \frac{\dot{B}_0 - \dot{B}_{\text{divide}}}{z_0 - z_{\text{divide}}}, \tag{3}$$

where $z_{divide}$ is the elevation at the ice divide (upper boundary of domain) and $z_0 = 0$ is the elevation at sea level. The anomalies at sea level ($\dot{B}_0 = +1$ m w.e.) and at the divide ($\dot{B}_{\text{divide}} = +0.2$ m w.e.) are not meant to mimic a specific climatic period. However, the resulting surface mass balance field resembles that of the pre-industrial climate, as simulated by Plach et al.[82]. We also did sensitivity tests with even stronger shifts of the surface mass balance (Section "Study limitations and future work"; Table S1).

The surface mass balance rate $\dot{B}$ is then calculated as

$$\dot{B} = \dot{B}_{MAR} + \Gamma z_s + \dot{B}_0, \tag{4}$$

where $\dot{B}_{MAR}$ is the present-day climatic surface mass balance from MAR, $z_s$ the ice-surface elevation and $\dot{B}_0$ is the surface mass balance anomaly at sea level.

*Ocean forcing.* Ocean forcing is parameterised based on observations of Petermann's ice shelf and modelling using the ocean model MITgcm[29,30]. Water properties in these simulations are consistent with measurements from moorings[23], as well as CTD (Conductivity, Temperature, Depth) casts from the Petermann 2015 Expedition[83]. In the model, we apply annual oceanic melt rates of the ice-shelf base. Melt rates vary linearly from zero at depths shallower than 200 m to 30 m/a at

depths deeper than 600 m (Fig. S2). Negligible seasonal variability of the deep-water properties occurs at Petermann[23], and our ocean boundary conditions are therefore robust as an annual forcing. Ocean undercutting (horizontal melt) at the calving front is only applied if the calving front is grounded. However, Petermann's ice shelf never disappears completely in our simulations, so grounded-front melt is never applied in practice.

In experiments where the oceanic melt is altered, oceanic melt rates are based on modelling experiments by[30]. Associated depth—melt rate profiles for a given temperature are given in Fig. S2. For recovery experiments, these profiles are shifted back to the present-day profile (Fig. S2).

## Data availability
Our work is based on numerical modelling and we provide the scripts necessary to reproduce our simulations (see Code Availability below). The underlying data used are publicly available and listed in Section S1.5 and Table S1.5 in the Supplementary Information.

## Code availability
The ISSM code is freely available from the ISSM website (https://issm.jpl.nasa.gov/download). Model scripts used to prepare and launch simulations are available at https://git.bolin.su.se/bolin/akesson-2022.

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

## Acknowledgements

The simulations were enabled by resources provided by the Swedish National Infrastructure for Computing (SNIC) at the National Supercomputer Centre (NSC), partially funded by the Swedish Research Council through grant agreements 2016-07213 and 2018-05973. The research was supported by the Swedish Research Council VR grants 2016-04021 (H.Å. and M.J.), 2018-04350 (C.S.) and 2020-05076 (J.N.). M.M. was supported by the Heising Simons Foundation grant 2019-1161 and 2021-3059. We would like to thank Andreas Plach for scientific advice on surface mass balance and the pre-industrial climate. Finally, Hamish Struthers at the National Computation Centre in Sweden is acknowledged for assistance concerning technical aspects in making the ISSM code run on the Tetralith high-performance cluster.

## Author contributions

The study was conceived by H.Å., M.M. and M.J. H.Å. prepared, ran and analysed the model simulations, with technical advice from M.M. and scientific input from M.M., M.J., C.S. and J.N. H.Å. made all figures except Fig. S1a, b (made by M.J.), and wrote the manuscript with significant contributions from M.M., C.S., J.N. and M.J.

## Funding

## Competing interests

The authors declare no competing interests.
