## [Peer Review File · Nature Communications]

Petermann Ice Shelf may not recover after a future breakupReviewers' Comments:

Reviewer #1:

Remarks to the Author:

Overall, this is a really neat paper that clearly addresses its research question. The methodology is sound and the numerical model used is proven in many papers. The authors are very thorough in the modelling experiments they conduct and, as a result, really get to the heart of the question. The question about what it takes to regrow an ice tongue is very interesting and an important one. As well as the comprehensive modelling, it is great to see the authors compare their work to the little data that are available to us.

My comments are relatively minor and in the attached document as in text notes. Most of these are tidying up / clarifying the argument, but also not conflating ice tongue and ice shelf, and with it the implication that results from Petermann are directly transferable e.g. to large Antarctic ice shelves. They may be, or they may not, but the work has not yet been done to confirm either way. As the authors note, this is an important area for future work, but I think you need to be careful about overreaching the applicability here.

Overall, this is a really nice manuscript, that makes an important contribution to knowledge and is very solid scientifically. My comments are really polishing. Thanks for an enjoyable read!

Reviewer #2:

Remarks to the Author:

This is a well-written manuscript that presents important results. While I believe that the approach used is sound, and the data valid, there are some aspects of the conclusions (or at least the way that they are written), that do not quite reflect the results presented. There are also some instances of conflated terminology (ice shelf vs ice tongue).

1. In the abstract and in the conclusions, I think it would be more accurate to state that the Petermann Gletscher ice shelf only recovers if ocean temperatures return to present conditions accompanied by increased melange and sea ice cover. I expect that these factors (ocean cooling and reduced calving) would likely act in tandem (and unlikely to act in isolation). i.e. cooler ocean temperatures would lead to more extensive melange and sea ice formation. Therefore, enforcing one without the other may be a little unrealistic? Presently I think the wording (mainly in the abstract and conclusion) seems somewhat pessimistic compared to the results presented.

2. Ice tongues have variously been defined as (or at least the terminology has been used in papers concerning) transient seasonal floating seaward extensions of glaciers (e.g. Amundson et al., (2010, doi:10.1029/2009JF001405); Moyer et al., (2019, doi: 10.1017/jog.2019.27), and topographically unconstrained floating seaward extensions of glaciers (e.g. DOI: <https://doi.org/10.1017/aog.2017.4>). Furthermore, Petermann Gletscher's floating extension has typically been referred to as an 'ice shelf' (e.g. Munchow et al., (2019, <https://doi.org/10.5670/oceanog.2016.101>)). I therefore think it would be better to use 'ice shelf' throughout the paper.

3. There are some instances where the manuscript's findings are suggested to be applicable to all ice shelves (e.g. line 43). Given that Petermann ice shelf is topographically constrained, and Petermann fjord has bathymetry that may differ from many other fjords, I'm not sure these inferences are justified.

Minor points by line number:

L5: Should be 'Paris Agreement'

L6: Is 'and glaciers' necessary?

L8-9: Consider changing 'can recover once future ice-shelf breakup and grounding-line retreat have unfolded' to 'can recover following future ice-shelf breakup and grounding-line retreat'

L10-15: This refers to main point 1 above.

L19: 'retreated and accelerated'?

L 20: Consider changing 'is known to have played a key role for this pattern' to 'is known to have played a key role in this behaviour'.

L22: Perhaps 'lower' could be 'limit'?

L23: I'm not sure 'back' is necessary?

L24: First mention of 'ice tongue' following use of 'ice shelf' in the Abstract. I think ice shelf should be used throughout the manuscript (see main point 2 above).

L29: Or at least the subsequent ice recession hides most sign of the earlier advance...

L34: 'sea level rise to be kept moderate'.

L38-39: The definition should be provided on the first use of 'ice tongue'. However, I don't think that ice tongue is the correct term (see main point 2).

L39: 'Petermann has lost'?

L43: In general? Not the specific case of a fjord-constrained ice shelf?

L44: 'testament' could be 'demonstrate'

Figure 1 caption: What do the years (colours) represent? It is not clear whether they are years from present or the induced warming or AD. From L54, I think the warming is induced in the year 2000? Perhaps this could be clarified and/or some more information added to the Figure 1 caption.

Figure 1 caption: For panel d, the caption states 'an ocean-warming reversal and a positive shift of the surface mass balance', while the figure annotation states 'ocean + atmospheric cooling'. I think these should be the same to avoid confusion.

L75: 'depend' should be 'depends'

L77: Here Fig 1e is referred to as representing 'increased net surface accumulation'. I think this should be consistent with the Fig 1 caption and annotation.

L96: I expect that these factors (ocean cooling and reduced calving) would likely act in tandem (and unlikely to act in isolation). i.e. cooler ocean temperatures would lead to more extensive melange and sea ice formation. Therefore, enforcing one without the other may be a little unrealistic? See main point 1.

Figure 3 caption: It would be useful to add annotation or a legend to explain the coloured lines. 'a ocean' should be 'an ocean'

L116-118: Somewhere in here I think a caveat along the lines of 'as long as there is sufficient ocean cooling' should be added. I also found section 5.1 less clearly written than the rest of the manuscript.

L133: Would a significant ocean cooling not be expected to lead to stronger melange and more extensive, thicker sea ice? See main point 1.

L135: Why would these effects (sea ice growth and damping of calving) not also occur during the ocean cooling after 2100 BP? Is it because that cooling was of insufficient magnitude or did not last for enough time? How do these inferred cooling events compare in magnitude and duration with that induced during the future experiments? I suppose I am wondering whether, based on the palaeo data, the enforced cooling would be expected by default to also cause increased sea ice formation and reduced calving.

L143: I wonder whether this overstates the extent and prevalence of melange a little? I think it would be more accurate to say 'melange is a dense mix of calved icebergs and sea ice that covers the inner part of many Greenlandic fjords.' Perhaps you could also say: 'Melange is more prevalent and extensive where glaciers have a high calving flux and in fjords that experience cooler ocean and atmospheric temperatures.' Or something along those lines.

L151: Is 'conversely' really appropriate here? the end effect (i.e. a melange strong enough to suppress calving) is the same.

L155: I'm not sure that your findings suggest this (because your model does not include sea ice growth). Your findings suggest that ice shelf recovery is facilitated by a higher calving stress threshold. You assume this is representative of a strong ice melange which could be related to more extensive and thicker sea ice cover.

L170: 'to fjord' should be 'the fjord' I think.

L173: 'to recover' could be 'in recovering'

L183: Could 'similarly' be 'relatively'? I don't think you have stated explicitly that the ocean forcing was simple.

L194: Is it worth saying a few words about your findings in relation to the potential differences

between a topographically constrained ice shelf such as that at Petermann, vs much larger laterally extensive ice shelves in Antarctica. For example, differences in the relative amount of backstress provided by ice mélange vs lateral traction?

L198: Should 'an' be 'the'? i.e. it is not immediately clear that your results for Petermann ice shelf would necessarily be applicable to all other ice shelves.

L199-200: But would this cooling ever occur in reality without a commensurate increase in mélange coverage and strength and more extensive sea ice? I suppose what I'm suggesting is that a combination of the direct and indirect effects of a reversal of ocean warming would be sufficient to allow the ice shelf to recover, and that separating these may be somewhat artificial.

Figure A1: It would be useful to adding text to identify the colours used for the grounding line in each set of experiments to the caption.

Figure A11: Might it be useful to have a separate arrow 'Time' pointing downwards to one side or the other of the figure?

Reviewer #3:

Remarks to the Author:

The Petermann glacier in North Greenland is of high interest, because it is one of very few marine glaciers in Greenland with a remaining floating ice tongue. The Petermann glacier has lost 40% of its tongue since 2008, and it is currently experiencing further retreat and thinning. The paper investigates the conditions needed for the Petermann glacier to grow back, if the tongue undergoes a complete break-up and is lost in a future warmer climate. This is a timely and relevant question to ask, and it contributes to the high attention on tipping points and irreversible changes in the ice mass loss in the public as well as the scientific community. As noted in the paper, it is in high demand to understand thresholds and tipping points related to the marine glaciers, and for this reason the paper is interesting for a wide audience.

The analysis itself is clearly presented in figures and text, generally well planned and interesting to read.

My main issue is related to the atmosphere forcing and the arguments for increasing smb in future colder climates. It makes sense to assume that smb increases near the margin in colder climates due to less runoff. But the accumulation rate in the interior should decrease in colder climates, following the relationship between temperature and accumulation rate found in ice cores (e.g. Dansgaard et al. 1993. Evidence for general instability of past climate from a 250-kyr ice-core record. *Nature*, Vol. 364, 218-220). The paper by Aschwanden et al. (2019) (referred to in line 245) also used a similar exponential scaling of precipitation with temperature. I suggest that the assumed anomaly is explained better to clarify these points, or perhaps reconsider. I don't expect that it would affect the results, since the atmospheric forcing is minor compared to calving, and because my comment only relates to the interior smb.

I have a some additional minor points:

1. It would be helpful with a more detailed discussion on the importance of the geometry of the fjord. Is the geometry controlling the stable positions, and does it make the results specific to Petermann, or are the results more general?
2. A few comments on the use of English: - a few sentences here and there needs to be checked. - Figure caption of fig. 1 is hard to read. Perhaps move the commas, so the sub-figures are listed like (a)....., (b) ... etc. - In the sentence lines 35-38, please insert "it" before "drains" and before "has one of the northern hemisphere's..."
3. I miss a more detailed discussion of the atmospheric forcing and the effect of surface meltwater on the floating tongue and calving rate. E.g. in section 2, lines 51 and 57. The effect of surface meltwater could be incorporated in the discussion of buttressing, sea ice and mélange as well, and would help argue for the assumed changes of the calving stress threshold.
4. I found the expression "before the Industrial Revolution" to be unusual and a little awkward. The usual expression is "in the pre-industrial period". Please consider to change.
5. The sentence in lines 200-202 sounds logically opposite to what it means to express. If ocean-

calving interactions are “fueled” by more extensive sea ice, it sounds like it strengthens, but instead it weakens. Perhaps change “fueled” to “suppressed”.

6. Figure A11 is a very nice cartoon!

7. In the discussion of the buttressing effects associated with a break-up of the tongue it is worthwhile to refer to the study by Rathmann et al. (2017) (Highly temporally resolved response to seasonal surface melt of the Zachariae and 79N outlet glaciers in northeast Greenland, *Geophys. Res. Lett.*, 44, doi:10.1002/2017GL074368). They showed that the outer >50 km of the floating tongue at 79th fjord did not have any significant buttressing effect and could be lost without affecting the flow. What are the conditions at Petermann today?

Authors: Responses to the Reviewers

Reviewer 1

Overall, this is a really neat paper that clearly addresses its research question. The methodology is sound and the numerical model used is proven in many papers. The authors are very thorough in the modelling experiments they conduct and, as a result, really get to the heart of the question. The question about what it takes to regrow an ice tongue is very interesting and an important one. As well as the comprehensive modelling, it is great to see the authors compare their work to the little data that are available to us.

My comments are relatively minor and in the attached document as in text notes. Most of these are tidying up / clarifying the argument, but also not conflating ice tongue and ice shelf, and with it the implication that results from Peterman are directly transferable e.g. to large Antarctic ice shelves. They may be, or they may not, but the work has not yet been done to confirm either way. As the authors note, this is an important area for future work, but I think you need to be careful about over reaching the applicability here.

This is a good point; we now further refine the wording, specifically in relation to unconfined ice shelves typically found in Antarctica, as also suggested by reviewer 2.

Overall, this is a really nice manuscript, that makes an important contribution to knowledge and is very solid scientifically. My comments are really polishing. Thanks for an enjoyable read!

We thank the reviewer for these encouraging words, and for taking the time to suggest concrete points of improvement. Our responses to the comments in-text are included in the attached PDF.

Reviewer 2

This is a well-written manuscript that presents important results. While I believe that the approach used is sound, and the data valid, there are some aspects of the conclusions (or at least the way that they are written), that do not quite reflect the results presented. There are also some instances of conflated terminology (ice shelf vs ice tongue).

1. In the abstract and in the conclusions, I think it would be more accurate to state that the Petermann Gletscher ice shelf only recovers if ocean temperatures return to present conditions accompanied by increased melange and sea ice cover. I expect that these factors (ocean cooling and reduced calving) would likely act in tandem (and unlikely to act in isolation). i.e. cooler ocean temperatures would lead to more extensive melange and sea ice formation. Therefore, enforcing one without the other

may be a little unrealistic? Presently I think the wording (mainly in the abstract and conclusion) seems somewhat pessimistic compared to the results presented.

Great point. When we started the study, we anticipated that a reversal of the ocean temperature (or at least an extreme scenario with zero basal melt) would allow the shelf to recover. It did not, and we found that calving prohibited recovery. Further experiments showed that recovery was possible if the calving stress threshold was increased sufficiently in the model's representation of calving. Further, we found a step-like response to changes in calving stress threshold at a critical value (Fig. A8). While there are uncertainties in the calving representation and how ice mélange may affect calving, the ocean-cooling experiment may be viewed as case where the mélange effects on calving are present but too weak to allow for a recovery of the ice shelf.

We agree that, in reality, a cooling ocean would likely allow mélange expansion and sea-ice growth. This is what we outline in Section 5.3 'Sea-ice induced advance of ice shelves' as a mechanism for recovery. In terms of our experimental setup, we however think that the ocean-cooling experiments is a clean way to isolate the oceanic influence vs other forcings/factors. We have also designed the ocean-cooling simulations as a simple reversal of the ocean warming imposed in Åkesson et al. 2021, as explained in Section 2, L58-60. Similarly, we performed 'Calving-only' experiments (see Table B1), to isolate the effect of calving. In a modelling framework we believe it is informative to do such isolated "sensitivity" simulations. For these reasons, we believe that the experimental setup should stay as it is. Moreover, it is unclear on what time scale and even which forcing (ocean or sea ice/mélange) that will change before the other (see also answer to L135 below).

In the revised manuscript, we have nevertheless changed the wording in the abstract and in the conclusions to address the concern of the reviewer. In the abstract, we have moderated to "Our experiments suggest that post-breakup recovery of *confined* ice shelves like Petermann's is unlikely, *unless iceberg calving rates are greatly reduced.*" (*added words in italics*).

In the conclusion, we moderate to "Our model simulations show that post-breakup recovery of an ice shelf is **extremely** difficult and requires a major climatic cooling to occur."

and

"Instead we highlight *sea-ice and ice mélange-induced suppression of calving, potentially accompanying colder ocean temperatures*, as a viable escape route from sustained mass loss and associated sea-level rise. *While the suppressive effect of sea ice on calving is supported by empirical evidence, it is not explicitly modelled and only implicitly accounted for in our calving parameterization.*"

We think that we already highlight that it is possible to avoid sustained mass loss by mentioning this "viable escape route". We however added the last sentence *in italics*

to highlight that while these mechanisms are supported by empirical evidence, they are not explicitly accounted for in our simulations.

2. Ice tongues have variously been defined as (or at least the terminology has been used in papers concerning) transient seasonal floating seaward extensions of glaciers (e.g. Amundson et al., (2010, doi:10.1029/2009JF001405); Moyer et al., (2019, doi: 10.1017/jog.2019.27), and topographically unconstrained floating seaward extensions of glaciers (e.g. DOI: <https://doi.org/10.1017/aog.2017.4>). Furthermore, Petermann Gletscher's floating extension has typically been referred to as an 'ice shelf' (e.g. Munchow et al., (2019, <https://doi.org/10.5670/oceanog.2016.101>)). I therefore think it would be better to use 'ice shelf' throughout the paper.

We agree that there was potential for confusion, and now use 'ice shelf' throughout. We also explain that Petermann is a topographically confined ice shelf, as opposed to the less confined ice shelves more common in Antarctica.

3. There are some instances where the manuscript's findings are suggested to be applicable to all ice shelves (e.g. line 43). Given that Petermann ice shelf is topographically constrained, and Petermann fjord has bathymetry that may differ from many other fjords, I'm not sure these inferences are justified

Good point; we now further refine this wording (e.g. line 43), also in light of the less confined ice shelves of Antarctica, as also suggested by reviewer 1.

Minor points by line number:

L5: Should be 'Paris Agreement'

Changed.

L6: Is 'and glaciers' necessary?

Well, we thought that the question of recovery is also an interesting one for glaciers without an ice shelf that has retreated (although this is not the main focus on our paper). So we keep this as is.

L8-9: Consider changing 'can recover once future ice-shelf breakup and grounding-line retreat have unfolded' to 'can recover following future ice-shelf breakup and grounding-line retreat'

Changed.

L10-15: This refers to main point 1 above.

See answer to main point 1 above.

L19: 'retreated and accelerated'?

Changed.

L 20: Consider changing 'is known to have played a key role for this pattern' to 'is known to have played a key role in this behaviour'.

Changed to "behaviour", and slightly rewritten due to suggestions from reviewer 1.

L22: Perhaps 'lower' could be 'limit'?

Changed to "reduce", to avoid word repetition later in the same sentence.

L23: I'm not sure 'back' is necessary?

Changed.

L24: First mention of 'ice tongue' following use of 'ice shelf' in the Abstract. I think ice shelf should be used throughout the manuscript (see main point 2 above).

Changed here and throughout, see above.

L29: Or at least the subsequent ice recession hides most sign of the earlier advance...

The reviewer is right, if the documentation consists of geomorphology then retreat can override the (few) traces of earlier advance. Ice-advances can also be captured based on sediment cores, of course. We have changed to "... but advancing glaciers generally leave fewer traces behind than those retreating, or the geomorphic imprints are overridden by subsequent ice recession."

L34: 'sea level rise to be kept moderate'.

Changed.

L38-39: The definition should be provided on the first use of 'ice tongue'. However, I don't think that ice tongue is the correct term (see main point 2).

Changed here and throughout, see above.

L39: 'Petermann has lost'?

Changed.

L43: In general? Not the specific case of a fjord-constrained ice shelf?

Changed to "...once a confined ice shelf has been lost"

L44: 'testament' could be 'demonstrate'

Changed.

Figure 1 caption: What do the years (colours) represent? It is not clear whether they are years from present or the induced warming or AD. From L54, I think the warming is induced in the year 2000? Perhaps this could be clarified and/or some more information added to the Figure 1 caption.

The years are AD, which has now been added to the colorbar title.

Figure 1 caption: For panel d, the caption states ‘an ocean-warming reversal and a positive shift of the surface mass balance’, while the figure annotation states ‘ocean + atmospheric cooling’. I think these should be the same to avoid confusion.

Changed to “increased surface mass balance (SMB)” in both caption and figure.

L75: ‘depend’ should be ‘depends’

Changed.

L77: Here Fig 1e is referred to as representing ‘increased net surface accumulation’. I think this should be consistent with the Fig 1 caption and annotation.

Changed to “increased surface mass balance (SMB)” in both caption and figure.

L96: I expect that these factors (ocean cooling and reduced calving) would likely act in tandem (and unlikely to act in isolation). i.e. cooler ocean temperatures would lead to more extensive melange and sea ice formation. Therefore, enforcing one without the other may be a little unrealistic? See main point 1.

The reviewer is right, however there are both experimental and scientific reasons to separate the forcings as well. See answer above to main point.

Figure 3 caption: It would be useful to add annotation or a legend to explain the coloured lines.

Done.

‘a ocean’ should be ‘an ocean’

Changed.

L116-118: Somewhere in here I think a caveat along the lines of ‘as long as there is sufficient ocean cooling’ should be added. I also found section 5.1 less clearly written than the rest of the manuscript.

We have added this caveat “as long as there is sufficient ocean cooling” and rewritten this section for clarify.

L133: Would a significant ocean cooling not be expected to lead to stronger melange and more extensive, thicker sea ice? See main point 1.

We agree that a uniform ocean cooling should lead to stronger ice mélange and thicker/more extensive sea ice. But the Arctic oceanography is characterized by a strong stratification with a cold upper halocline, protecting the sea ice from a warm intermediate water mass, in these areas having an Atlantic origin. Thus, an ocean cooling deeper down below the halocline, for example through reduced influx of Atlantic water, will greatly affect ice shelf melting, but not necessarily formation of sea ice. The sea ice will rather be influenced by local atmospheric cooling and fresh water influx. This kind of complex oceanography may play a large role in why oceanic temperatures and sea ice are not always varying in sync.

We now explain these aspects at the end of Section 4.

L135: Why would these effects (sea ice growth and damping of calving) not also occur during the ocean cooling after 2100 BP? Is it because that cooling was of insufficient magnitude or did not last for enough time? How do these inferred cooling events compare in magnitude and duration with that induced during the future experiments? I suppose I am wondering whether, based on the palaeo data, the enforced cooling would be expected by default to also cause increased sea ice formation and reduced calving.

Yes possibly, but it is not clear from the reconstructions that both ocean cooling and sea-ice expansion occurred from 2100 to 1400 BP. Most likely there is some lag in the system, delaying sea-ice growth/landfast ice.

An ocean cooling and sea-ice growth do not necessarily vary in tandem in the real world. An 'ocean cooling' in our experiments means that sub-ice shelf melt is reduced (cf. Section A3.2). In reality, sub-shelf melt is mainly driven by available subsurface ocean heat which, in this case, is controlled by remote oceanographic conditions rather than local forcing. In contrast, local atmospheric conditions govern surface waters and sea-ice conditions. Indeed, the ocean - sea ice - atmosphere is a coupled system, but changes are likely to be asynchronous and to occur on different time scales. Specifically, a subsurface ocean cooling should in general favour sea-ice and ice mélange growth, but the time lag could be significant. These aspects are why we treated ocean cooling and reduced calving separately in our experiments.

L143: I wonder whether this overstates the extent and prevalence of mélange a little? I think it would be more accurate to say 'mélange is a dense mix of calved icebergs and sea ice that covers the inner part of many Greenlandic fjords.' Perhaps you could also say: 'Mélange is more prevalent and extensive where glaciers have a high calving flux and in fjords that experience cooler ocean and atmospheric temperatures.' Or something along those lines.

We now change both sentences as the reviewer suggested, just with a slight modification:

“Mélange tends to be more prevalent and extensive in front of glaciers with vigorous calving, which usually occurs in warmer fjords. Conversely, in fjords with cooler ocean and atmospheric temperatures, calving rates are expected to be lower, but calved icebergs are subject to less efficient melt, favouring a longer-lasting mélange.”

L151: Is 'conversely' really appropriate here? the end effect (i.e. a mélange strong enough to suppress calving) is the same.

Yes this wasn't clear, as also pointed out by reviewer 1. We now clarify that the end effect is the same, but the mechanisms are different.

L155: I'm not sure that your findings suggest this (because your model does not include sea ice growth). Your findings suggest that ice shelf recovery is facilitated by a higher calving stress threshold. You assume this is representative of a strong ice mélange which could be related to more extensive and thicker sea ice cover.

This indeed is more nuanced, thanks. We change to: “For Petermann our findings suggest that in a cooling climate, ice-mélange induced suppression of calving can facilitate ice-shelf recovery, an effect which in our simulations is represented as a higher calving stress threshold (Section 4 and A2). We postulate that shorter calving seasons and a stronger ice mélange, both promoted by sea-ice growth, are the key physical mechanisms for ice-shelf recovery that has occurred in the past and may occur in the future.”

L170: ‘to fjord’ should be ‘the fjord’ I think.

Changed.

L173: ‘to recover’ could be 'in recovering'

Changed.

L183: Could ‘similarly’ be ‘relatively’? I don't think you have stated explicitly that the ocean forcing was simple.

Changed.

L194: Is it worth saying a few words about your findings in relation to the potential differences between a topographically constrained ice shelf such as that at Petermann, vs much larger laterally extensive ice shelves in Antarctica. For example, differences in the relative amount of backstress provided by ice mélange vs lateral traction?

Yes, this a very good and interesting point, also raised by the other reviewers. The general notion is that laterally extensive, unconfined ice shelves in Antarctica provide no backstress/buttressing to upstream ice. We searched a bit in the literature; a recent study by Wearing et al. (2020) investigates buttressing from unconfined ice shelves in detail and conclude that “most unconfined ice shelves provide insignificant buttressing

today” because they are extensively damaged, resulting in a low effective viscosity. They find that ice shelves like Thwaites consist of very weak ice. However, they note that where ice shelves are stabilised by sea ice (Mertz Ice Shelf), the effective viscosity is apparently higher due to less fractured ice. The higher viscosity enables these shelves to provide buttressing through hoop-stresses (stresses perpendicular to the ice-shelf radial spreading). They even hypothesize that “sea ice may prevent an ice tongue from collapsing and potentially allows it to advance.” which is essentially what we postulate based on our experiments. Moreover, Massom et al. (2018) show a strong temporal coincidence between reduced sea ice offshore of Antarctic ice shelves (Wilkins and Larsen) and disintegration. For Wilkins ice shelf they also illustrate a “strong temporal coincidence” between ice-shelf collapse and breakup/absence of landfast sea ice, as well as the converse situation (heavy pack ice and no disintegration).

At face value, the insights from Wearing et al and Massom et al suggest that our findings for Petermann could be transferable to Antarctic ice shelves, although this remains to be directly tested (as already pointed out on L195).

We now discuss these aspects towards the end of Section 6.

When it comes to buttressing from ice melange and lateral traction, we are not aware of studies comparing these two directly. We expect that lateral drag as a resistive stress would have a stronger direct effect on ice flow, while ice melange rather influences calving rates, which in turn of course can affect upstream ice dynamics.

L198: Should ‘an’ be ‘the’? i.e. it is not immediately clear that your results for Petermann ice shelf would necessarily be applicable to all other ice shelves.

Changed.

L199-200: But would this cooling ever occur in reality without a commensurate increase in melange coverage and strength and more extensive sea ice? I suppose what I'm suggesting is that a combination of the direct and indirect effects of a reversal of ocean warming would be sufficient to allow the ice shelf to recover, and that separating these may be somewhat artificial.

Possibly yes, but this becomes a bit convoluted since the future scenario (Åkesson et al. 2021) did not account for indirect effects such as calving and sea ice. To enable a fair comparison, these indirect effects would have to be included in both the retreat and advance simulations, which they are not. See also answer to L135 above.

Figure A1: It would be useful to adding text to identify the colours used for the grounding line in each set of experiments to the caption.

We have now added an explanation.

Figure A11: Might it be useful to have a separate arrow 'Time' pointing downwards to one side or the other of the figure?

We were tempted to do this, but this would somewhat contradict the direction of the “recovery” arrow, from “post-Paris” to “stability”.

References

Massom, R. A., Scambos, T. A., Bennetts, L. G., Reid, P., Squire, V. A., & Stammerjohn, S. E. (2018). Antarctic ice shelf disintegration triggered by sea ice loss and ocean swell. *Nature*, 558(7710), 383-389.

Wearing MG, Kingslake J, Worster MG (2020). Can unconfined ice shelves provide buttressing via hoop stresses? *Journal of Glaciology* 66(257), 349–361.
<https://doi.org/10.1017/jog.2019.101>

Åkesson, H., Morlighem, M., O'Regan, M., & Jakobsson, M. (2021). Future projections of Petermann Glacier under ocean warming depend strongly on friction law. *Journal of Geophysical Research: Earth Surface*, e2020JF005921.

Reviewer #3 (Remarks to the Author):

The Petermann glacier in North Greenland is of high interest, because it is one of very few marine glaciers in Greenland with a remaining floating ice tongue. The Petermann glacier has lost 40% of its tongue since 2008, and it is currently experiencing further retreat and thinning. The paper investigates the conditions needed for the Petermann glacier to grow back, if the tongue undergoes a complete break-up and is lost in a future warmer climate. This is a timely and relevant question to ask, and it contributes to the high attention on tipping points and irreversible changes in the ice mass loss in the public as well as the scientific community. As noted in the paper, it is in high demand to understand thresholds and tipping points related to the marine glaciers, and for this reason the paper is interesting for a wide audience.

The analysis itself is clearly presented in figures and text, generally well planned and interesting to read.

My main issue is related to the atmosphere forcing and the arguments for increasing smb in future colder climates. It makes sense to assume that smb increases near the margin in colder climates due to less runoff. But the accumulation rate in the interior should decrease in colder climates, following the relationship between temperature and accumulation rate found in ice cores (e.g. Dansgaard et al. 1993. Evidence for

general instability of past climate from a 250-kyr ice-core record. *Nature*, Vol. 364, 218-220). The paper by Aschwanden et al. (2019) (referred to in line 245) also used a similar exponential scaling of precipitation with temperature. I suggest that the assumed anomaly is explained better to clarify these points, or perhaps reconsider. I don't expect that it would affect the results, since the atmospheric forcing is minor compared to calving, and because my comment only relates to the interior smb.

We would like to thank the reviewer for these very encouraging words and for sharing insights into the atmospheric side of the surface mass balance in particular. The reviewer is completely right that, in a colder climate, there is most likely a “desertification effect” (less snowfall) over the Greenland Ice Sheet, which should give lower accumulation rates compared to the present climate. Colder climatic conditions will also decrease the ablation, and the net change in mass balance will be determined by competing changes in accumulation and ablation. As discussed by the reviewer (and also appendix A3.1), on high parts of an ice sheet with very low or even no ablation, a cooling tends to make the net surface mass balance less positive. At lower altitudes, with stronger ablation, a cooling tends to make the net surface mass balance less negative. A cooling (warming) thus generally entails spatial changes of the surface mass balance, but the integrated effect on the mass balance generally causes ice-sheet growth (retreat). As they stand, the surface mass balance in the “atmospheric cooling”-experiments is perturbed relative to the present-day surface mass balance with +1 m w.e. at sea level, +0.2 m w.e. at the divide, varying linearly in between. Following the reviewer's comments, a more physical scenario would be to switch to a negative perturbation at the divide, for example -0.2 m w.e. (but still with a net mass balance ≥ 0 at divide).

However, we would like to keep the simulations as they stand, but rephrase what they mean. The advantage of this is that the current experiments test the combined effect of (strongly) reduced ablation and (slightly) increased accumulation (albeit we do not expect that both these occur at the same time with a cooler atmosphere). To be clear, we renamed the former “atmosphere cooling” throughout the manuscript to “increased surface mass balance (SMB)”, abbreviated “SMB” in some figures due to space constraints. We also clarify the rationale in Section 3 and A3.1.

I have a some additional minor points:

1. It would be helpful with a more detailed discussion on the importance of the geometry of the fjord. Is the geometry controlling the stable positions, and does it make the results specific to Petermann, or are the results more general?

Overall, we think that the major control of recovery is the ice-shelf dynamics and calving regime, but with local impacts of the geometry. We now discuss this in more

detail in Section 5.1. See also new discussion about confined vs unconfined ice shelves towards end of Section 5.4.

2. A few comments on the use of English:
a few sentences here and there needs to be checked.

We gave the manuscript another proof-read.

Figure caption of fig. 1 is hard to read. Perhaps move the commas, so the sub-figures are listed like (a)....., (b) ... etc.

Done.

In the sentence lines 35-38, please inserts “it” before “drains” and before “has one of the northern hemisphere’s...”

We do not think adding “it” in these cases is necessary.

3. I miss a more detailed discussion of the atmospheric forcing and the effect of surface meltwater on the floating tongue and calving rate. E.g. in section 2, lines 51 and 57. The effect of surface meltwater could be incorporated in the discussion of buttressing, sea ice and mélange as well, and would help argue for the assumed changes of the calving stress threshold.

While there are abundant surface lakes on the present-day ice shelf, Petermann is not thought to be as susceptible to breakup due to hydrofracture compared to some Antarctic ice shelves, like Larsen B in 2002 (Macdonald et al. 2018). We however now mention the potential influence of supraglacial water and links to ice-shelf recovery in the discussion, Section 5.4, where appropriate.

4. I found the expression “before the Industrial Revolution” to be unusual and a little awkward. The usual expression is “in the pre-industrial period”. Please consider to change.

Changed.

5. The sentence in lines 200-202 sounds logically opposite to what it means to express. If ocean-calving interactions are “fueled” by more extensive sea ice, it sounds like it strengthens, but instead it weakens. Perhaps change “fueled” to “suppressed”.

Changed, see also response to reviewer 2 above.

6. Figure A11 is a very nice cartoon!

Thank you!

7. In the discussion of the buttressing effects associated with a break-up of the tongue

it is worthwhile to refer to the study by Rathmann et al. (2017) (Highly temporally resolved response to seasonal surface melt of the Zachariae and 79N outlet glaciers in northeast Greenland, *Geophys. Res. Lett.*, 44, doi:10.1002/2017GL074368). They showed that the outer >50 km of the floating tongue at 79th fjord did not have any significant buttressing effect and could be lost without affecting the flow. What are the conditions at Petermann today?

Hill et al (2018) did a similar study for Petermann. They artificially removed successive parts of Petermann's ice shelf, and looked at the instant velocity response. They found that the outer ~60 km (>12 km from the grounding line) of the ice shelf appears to be passive, providing little buttressing. While interesting, a caveat with these types of "instant-response studies" done by Rathmann et al and Hill et al is that we do not know what the transient effects are, including positive feedbacks related to calving and subshelf melt.

We now briefly discuss these aspects in Section 5.1, citing Rathmann et al and Hill et al 2018.

References

Hill, E. A., Gudmundsson, G. H., Carr, J. R., & Stokes, C. R. (2018). Velocity response of Petermann Glacier, northwest Greenland, to past and future calving events. *The Cryosphere*, 12(12), 3907-3921.

Macdonald, G. J., Banwell, A. F., & MacAYEAL, D. R. (2018). Seasonal evolution of supraglacial lakes on a floating ice tongue, Petermann Glacier, Greenland. *Annals of Glaciology*, 59(76pt1), 56-65.

Rathmann, N. M., Hvidberg, C. S., Solgaard, A. M., Grinsted, A., Gudmundsson, G. H., Langen, P. L., ... & Kusk, A. (2017). Highly temporally resolved response to seasonal surface melt of the Zachariae and 79N outlet glaciers in northeast Greenland. *Geophysical Research Letters*, 44(19), 9805-9814.

Åkesson, H., Morlighem, M., O'Regan, M., & Jakobsson, M. (2021). Future projections of Petermann Glacier under ocean warming depend strongly on friction law. *Journal of Geophysical Research: Earth Surface*, e2020JF005921.

Reviewers' Comments:

Reviewer #1:

Remarks to the Author:

My initial comments on the manuscript were minor and the authors have done an excellent job of addressing them. I have no further comments / revisions. The paper is interesting and will make an important contribution to the field. The figures are excellent and the text is comprehensive, fluent and easy to read. Overall, an excellent paper, which I very much enjoyed reading.

Reviewer #2:

Remarks to the Author:

The authors have done a great job incorporating or rebutting my suggestions. I have just a few remaining minor typographical suggestions:

L5: Should you use 'grounded-ice' to be consistent with its use earlier in the summary? I suppose that 'inland' ice may not necessarily be grounded? You may also wish to check this throughout the manuscript.

L27: 'play' should be 'plays'

L99: 'calving rates alone do' (or 'reduced calving alone does')

L240: Should you use all capitals for MICI (or maybe none as you then do not use the acronym)?

Reviewer #3:

Remarks to the Author:

The authors have responded satisfactorily to all my comments. I agree with the authors that the simulations can be kept as they are, and the points are sufficiently addressed by clarifying the text and explaining better the simulations. I also find that comments from the other reviewers are addressed well, and the discussion in general is clarified and appears complete and well written. I have no further comments.

Authors: Responses to the Reviewers

Reviewer 1

My initial comments on the manuscript were minor and the authors have done an excellent job of addressing them. I have no further comments / revisions. The paper is interesting and will make an important contribution to the field. The figures are excellent and the text is comprehensive, fluent and easy to read. Overall, an excellent paper, which I very much enjoyed reading.

We thank the reviewer for their time and insight that has significantly improved the paper.

Reviewer 2

The authors have done a great job incorporating or rebutting my suggestions. I have just a few remaining minor typographical suggestions:

L5: Should you use 'grounded-ice' to be consistent with its use earlier in the summary? I suppose that 'inland' ice may not necessarily be grounded? You may also wish to check this throughout the manuscript.

Changed. Also changed on L42.

L27: 'play' should be 'plays'

Done.

L99: 'calving rates alone do' (or 'reduced calving alone does')

Done.

L240: Should you use all capitals for MICI (or maybe none as you then do not use the acronym)?

Removed the capitalisation in Marine, since we do not use the acronym.

Finally, we would like to thank the reviewer for fruitful discussions and constructive feedback.

Reviewer 3

The authors have responded satisfactorily to all my comments. I agree with the authors that the simulations can be kept as they are, and the points are sufficiently addressed by clarifying the text and explaining better the simulations. I also find that comments from the other reviewers are addressed well, and the discussion in general is clarified and appears complete and well written. I have no further comments.

We thank the reviewer for the discussions and great advice on how to improve the manuscript.